# Representation Learning for Integrating Multi-domain Outcomes to Optimize Individualized Treatments

**Yuan Chen**
Department of Biostatistics
Columbia University
New York, NY 10032
yc3281@cumc.columbia.edu

**Donglin Zeng**
Department of Biostatistics
University of North Carolina at Chapel Hill
Chapel Hill, NC 27516
dzeng@email.unc.edu

**Tianchen Xu**
Department of Biostatistics
Columbia University
New York, NY 10032
tx2155@cumc.columbia.edu

**Yuanjia Wang**
Department of Biostatistics
Columbia University
New York, NY 10032
yw2016@cumc.columbia.edu

## Abstract

For mental disorders, patients' underlying mental states are non-observed latent constructs which have to be inferred from observed multi-domain measurements such as diagnostic symptoms and patient functioning scores. Additionally, substantial heterogeneity in the disease diagnosis between patients needs to be addressed for optimizing individualized treatment policy in order to achieve precision medicine. To address these challenges, we propose an integrated learning framework that can simultaneously learn patients' underlying mental states and recommend optimal treatments for each individual. This learning framework is based on the measurement theory in psychiatry for modeling multiple disease diagnostic measures as arising from the underlying causes (true mental states). It allows incorporation of the multivariate pre- and post-treatment outcomes as well as biological measures while preserving the invariant structure for representing patients' latent mental states. A multi-layer neural network is used to allow complex treatment effect heterogeneity. Optimal treatment policy can be inferred for future patients by comparing their potential mental states under different treatments given the observed multi-domain pre-treatment measurements. Experiments on simulated data and a real-world clinical trial data show that the learned treatment polices compare favorably to alternative methods on heterogeneous treatment effects, and have broad utilities which lead to better patient outcomes on multiple domains.

## 1 Introduction

For mental disorders, patients' mental states are of primary interest and are crucial for optimizing individualized treatment policy. However, a patient's true mental state is not observed due to the complex nature of the disease. We only have assessments from disease diagnostic symptoms in an instrument or questionnaire that consists of multiple items (an item is a variable on an instrument/questionnaire). For example, the Hamilton Depression Rating Scale (HAM-D, [6]), which consists of 17 diagnostic symptom items, is designed to measure a patient's depression severity. Therefore, a patient's true underlying mental status needs to be inferred based on the the observed multi-domain measurements in order to prescribe optimal treatments and achieve precision medicine.

Current methods for learning individualized treatment rules (ITRs) focus on optimizing a single observed outcome variable. For example, the sum score of the 17 symptom items in HAM-D has been used as the outcome variable for learning ITRs for treating depression. However, there are some limitations for using such a simple summary score. First, questions in HAM-D such as "insomnia: early in the night", "insomnia: middle of the night", "insomnia: early morning", "suicidal idea", and "loss of weight", etc., are only partial reflections of a patient's true underlying mental states. Furthermore, they are not equal reflective measures of depression [5]. Second, several items overlap and tap on similar construct (e.g., 3 items about insomnia in HAM-D). Taking a simple sum of the symptom scores does not differentiate the items. Therefore, a simple summary score of the observed measurements can not give a full and accurate assessment of the disorder and may not serve as an adequate representation of a patient's true underlying mental states for treatment recommendation.

On the other hand, in psychiatry and psychology, measurement models (e.g., factor analysis models, item response theory models) have a long history of using latent variables to capture the underlying constructs that describe relations among a class of events or variables that share common disease causes [4]. The observable phenomena (e.g., psychological measurements) are influenced by the underlying shared causes/constructs (e.g., true mental functions), and these underlying constructs explains the correlations among the observed measures. By modeling the lower-dimensional underlying latent constructs, the noise in the observed symptoms (e.g., weight loss not due to depression) can be eliminated while the meaningful latent constructs (e.g., underlying mental functions) are manifested.

Therefore, leveraging the measurement model theory, we propose an integrated framework for representing patients' latent mental states based on the observed indicative psychological measurements while learning optimal treatments that maximize individuals' underlying mental states (graphical representation shown in Figure 1.a). The proposed model incorporates observed multi-domain measurements from both pre- and post-treatment phases, while the invariant representation structure is preserved guided by measurement model theory. A multi-layer neural network is used to model the complex treatment effect mechanism between the latent mental states and patients' biological factors. Optimal treatment policy can be inferred for future patients based on their pre-treatment measurements by comparing the potential mental states under different treatments estimated from the model. The proposed model utilizes data from randomized controlled trials (RCTs), which is considered as the "gold standard" for comparing treatment effects since treatment assignments are randomized and thus free from unobserved confounding. Using data from RCTs, we can learn ITRs with causal interpretations, and we describe the details in section 3. However, our method can easily be modified to learn ITRs from observational data, and we discuss it in section 5.

## 2 Related work

Machine learning methods and neural networks have been adapted to address patients' heterogeneous treatment responses for optimal treatment recommendations. Causal forest [29] was proposed to estimate the treatment effect for leaf-wise subgroups. TARNet [23] uses a neural net to learn a representation of the feature variables, relies on which the potential outcomes under different treatments are constructed. Recent papers [16, 1, 2, 24] build neural nets to alleviate confounding effects from observational data to learn individual treatment effect (ITE).

Instead of estimating ITEs to infer optimal individualized treatments, learning algorithms for ITRs have been proposed by directly or indirectly maximizing the value function. Value function is a common metric to evaluate treatment rules [20] and policies in reinforcement learning [7]. Value function of an ITR $d$ is defined as

$$\mathcal{V}(d) = E^d[R] = E[R(d)], \tag{1}$$

where $R(d)$ is the potential outcome under the treatment assigned by ITR $d$. O-learning [31, 32] and its robust version [14] maximize the value function by solving a weighted classification problem. Methods that indirectly maximize the value function through regression models of the outcomes or contrasts of outcomes under different treatments include regression-based Q-learning [30, 20], A-learning [19, 26], and G-computation [10, 18]. Recent work have incorporated neural nets to these methods to allow more flexible model structures. [13] uses neural nets to construct the "Q-function" for the potential outcomes (we reference this method as "deep-Q"). [11] uses a CNN to model the contrast/regret function in "A-learning" (we reference this method as "deep-A"). [17] adopts neural nets to construct the outcome residuals and the weighted classification model in O-learning (we reference this method as "deep-O").

The above methods all focus on optimizing a single observed outcome, and as discussed in section 1, they are not ideal for mental disorders where the observed outcomes are multivariate and the outcomes of interest (the true mental states) are not observed. Our approach addresses this gap by integrating multi-domain measures to learn patients' latent mental states according to measurement model theories, based on which ITRs are learned to maximize the value function in terms of patients' overall mental health. Our approach is fundamentally different from other latent variable modeling approach in that we combine measurement models and ITR learning under an unified loss where the latent states and ITRs are simultaneously learned with an iterative procedure. Compared to a straight-forward two-step procedure, our method is more efficient borrowing strength from data before and after treatment with shared representation parameters while preserving the consistency of the latent constructs. Additionally, traditional measurement models cannot deal with mixed-type input variables. For example, factor analysis typically deals with continuous measures and item response theory model works for binary measures. By utilizing the neural network learning structure, we can effectively incorporate mixed-type inputs without complicating the model much. The general framework and the model details are presented in section 3.

## 3  Method

Suppose $K$ unobserved binary latent variables ($K$ is known) are used to characterize a patient's latent mental state (e.g., positive mood versus negative mood). In practice, $K$ can be chosen based on scientific knowledge or by tuning as a hyper-parameter. We adopt the potential outcomes framework to introduce the notations and results. Let $Z_{0k} \in \{0, 1\}$ denote the $k$th latent variable at the baseline, and let $Z_{1k}^{(a)} \in \{0, 1\}$ denote the corresponding potential latent variable after being treated with treatment $a$. Further denote $\mathbf{Z}_0 = (Z_{01}, \cdots, Z_{0K})^T$ and $\mathbf{Z}_1^{(a)} = (Z_{11}^{(a)}, \cdots, Z_{1K}^{(a)})^T$. For the ease of illustration, we consider binary treatment decisions with $a \in \{-1, 1\}$. Our method can be easily extended to handle multiple treatment choices.

### 3.1  Individualized treatment rules (ITRs) with latent outcomes

Our goal is to maximize patients' overall mental health state by prescribing different treatment $a$ for different patients depending on each patient's individualized baseline mental states $\mathbf{Z}_0$ and other baseline characteristics $\mathbf{X}$ (e.g., demographics, biological measures). Treatment prescription can be determined by a decision function that maps a patient's feature space (both observed and unobserved), $(\mathbf{Z}_0, \mathbf{X})$, to the domain of treatments $a \in \{-1, 1\}$. A patient's overall mental health state post treatment is represented by a known aggregate function of $\mathbf{Z}_1^{(a)}$, which we denote as $g(\mathbf{Z}_1^{(a)})$. The choice of $g(\cdot)$ may depend on application but a common choice is the total sum similar to aggregating sub-scales of instruments commonly used in psychiatry [21]. The optimal ITR, denoted as $d^*$, is then an ITR that maximizes a patient's post-treatment mental state given by $d^*(\mathbf{X}, \mathbf{Z}_0) = \operatorname{argmax}_a E[g(\mathbf{Z}_1^{(a)}) | \mathbf{X}, \mathbf{Z}_0]$.

Since $\mathbf{Z}_0$ and $\mathbf{Z}_1^{(a)}$ are unobserved, to learn optimal ITRs we need to infer patients' mental states based on their observed psychological measurements that are reflective of their mental status, sometimes called manifest indicators [4, 12]. These informative measurements include items collected from multiple domains, for example, depression symptoms on the HAM-D rating scale and patient functioning collected on Work and Social Adjustment Scale (WSAS). Specifically, let $\mathbf{Y}_0$ denote the baseline observed indicators for $\mathbf{Z}_0$, and let $\mathbf{Y}_1^{(a)}$ denote the corresponding potential indicators for $\mathbf{Z}_1^{(a)}$ after being treated by treatment $a$. Assume there are $J$ relevant measures denoted by $\mathbf{Y}_0 = (Y_{01}, \cdots Y_{0J})^T$ and $\mathbf{Y}_1^{(a)} = (Y_{11}^{(a)}, \cdots, Y_{1J}^{(a)})^T$. For many mental disorders, multiple items are used to measure each domain of the disease, and thus the number of necessary latent mental states $K$ is often much smaller than the number of items $J$ [27, 28, 25].

We present the relationships of these variables in Figure 1.a, where the variables in squared boxes are observed and those in circles are latent. Essentially, treatment changes a patient's underlying mental health status from $\mathbf{Z}_0$ to $\mathbf{Z}_1^{(a)}$, which is indicated by the changes in the observed psychological measurements (from $\mathbf{Y}_0$ and $\mathbf{Y}_1^{(a)}$). The treatment effect depends on or is modified by the patient's baseline mental status $\mathbf{Z}_0$ and other baseline characteristics $\mathbf{X}$. That's the reason why we have heterogeneous treatment responses between individuals.

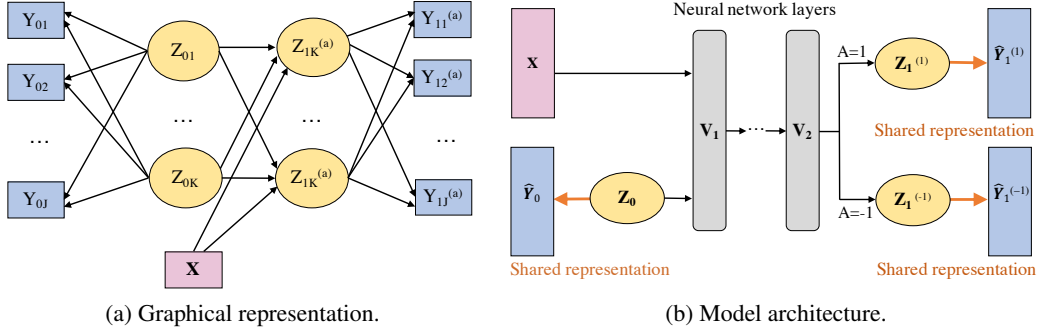

(a) Graphical representation.  (b) Model architecture.

Figure 1: (a) Graphical representation of the latent mental states and observed multi-domain symptom measures based on measurement theory. We note that $\mathbf{X}$ and $\mathbf{Z}_0$ can be correlated. (b) Model architecture for learning optimal ITRs. Paths highlighted in bold and color share the same parameters. $\widehat{\mathbf{Y}}_0, \widehat{\mathbf{Y}}_1^{(1)}, \widehat{\mathbf{Y}}_1^{(-1)}$: the model estimate for $\mathbf{Y}_0, \mathbf{Y}_1^{(1)}, \mathbf{Y}_1^{(-1)}$, respectively.

Furthermore, according to the measurement theory, for reliable instruments, the relationship between the observed items and the underlying latent constructs are assumed to be stable over different phases [15]. Thus, we assume the conditional distribution of $\mathbf{Y}_0$ given $\mathbf{Z}_0$ is the same as the conditional distribution of $\mathbf{Y}_1^{(a)}$ given $\mathbf{Z}_1^{(a)}$. In other words, how the items are measured and how they reflect the underlying disease status do not change over time. This preserves the validity of the latent variables.

## 3.2  Model for learning optimal ITRs

We assume that data are collected from a randomized trial and consist of the observations $(\mathbf{Y}_{i0}, \mathbf{X}_i, A_i, \mathbf{Y}_{i1})$, $i = 1, ..., n$, from $n$ independent patients. For subject $i$, $A_i$ denotes the treatment assignment, and we allow the treatment randomization probability to depend on $\mathbf{X}_i$. Because of randomization, $A_i$ is independent of the potential outcomes $\mathbf{Z}_{i1}^{(a)}$ and $\mathbf{Y}_{i1}^{(a)}$ conditional on $\mathbf{X}_i$. We assume the standard stable unit treatment value assumption (SUTVA) holds, i.e., $\mathbf{Z}_{i1}^{(a)} = \mathbf{Z}_{i1}$ and $\mathbf{Y}_{i1} = \mathbf{Y}_{i1}^{(a)}$ when $A_i = a$, where $\mathbf{Z}_{i1}$ is the true post-treatment mental states for subject $i$. Thus, due to randomization of treatments, we obtain $E[g(\mathbf{Z}_1^{(a)})|\mathbf{X}, \mathbf{Z}_0] = E[g(\mathbf{Z}_1^{(a)})|\mathbf{X}, \mathbf{Z}_0, A = a] = E[g(\mathbf{Z}_1)|\mathbf{X}, \mathbf{Z}_0, A = a]$. Because the optimal ITR maximizes post-treatment overall mental states, it can be obtained by $d^*(\mathbf{X}, \mathbf{Z}_0) = \operatorname{argmax}_a E[g(\mathbf{Z}_1)|\mathbf{X}, \mathbf{Z}_0, A = a]$.

**Model architecture**  Leveraging the structural representation in Figure 1.a, we propose a neural network to learn the latent representations $\mathbf{Z}$'s from multiple observed items in $\mathbf{Y}$ from both pre- and post-treatment phases. The computational architecture of our method is shown in Figure 1.b. We introduce a multi-layer neural network (represented by $\mathbf{V}_1$ to $\mathbf{V}_2$ in Figure 1.b) to capture the complex association among disease symptom measures and biological factors, in order to allow complex interactions with the treatment. Common structure through $\mathbf{V}_1$ to $\mathbf{V}_2$ across different treatment groups is used to capture the shared baseline associations between variables, and also we borrow strength for learning. We fit model for $\mathbf{Z}^{(1)}$ and $\mathbf{Z}^{(-1)}$ separately ($\mathbf{V}_2$ to $\mathbf{Z}_1^{(a)}$) instead of including observed treatment as an input variable as part of $\mathbf{X}$, because the treatment effect mechanism on the latent mental states can be potentially different between treatment groups. Furthermore, by the invariant structure between latent mental states and observed measurements as discussed in section 3.1, the paths from $\mathbf{Z}_0$ to $\mathbf{Y}_0$ and $\mathbf{Z}_1^{(a)}$ to $\mathbf{Y}_1^{(a)}$ across treatment groups share the same parameters (highlighted colored paths in Figure 1.b), and the representation structure is described below.

**Representation structure and interpretation**  We assume each discrete-valued $Y_{0j}$ given $\mathbf{Z}_0$ (same for $Y_{1j}^{(a)}$ given $\mathbf{Z}_1^{(a)}$) follows a multinomial distribution, and the representation structure between $\mathbf{Z}_0$ and discrete-valued $Y_{0j}$ (same for $\mathbf{Z}_1$ and $Y_{1j}$) is given by $P(Y_{0j} = m|\mathbf{Z}_0) = \exp(\alpha_{jm} + \sum_k \beta_{jkm} Z_{0k}) / \sum_{p=0}^{l_j} \exp(\alpha_{jp} + \sum_k \beta_{jkp} Z_{0k})$, for $m = 0, ..., l_j$. We denote $\boldsymbol{\beta}_{jk} = (\beta_{jk0}, ..., \beta_{jkl_j})^T$, and it reflects the correlation between discrete-valued item $j$ and latent domain $k$. Specifically, if $\beta_{jk0} < ... < \beta_{jkl_j}$ then $Y_{0j}$ (or $Y_{1j}$) is positively correlated with $Z_{0k}$

(or $Z_{1j}$), and if $\beta_{jk0} > ... > \beta_{jkl_j}$ then $Y_{0j}$ (or $Y_{1j}$) is negatively correlated with $Z_{0k}$. For continuous $Y_{0j}$, we model it with $\alpha_j + \sum_k \beta_{jk} Z_{0k}$, and then $\beta_{jk}$ itself indicates the association between item $j$ and latent domain $k$. To achieve these representation structures, softmax activation is applied to discrete-valued $Y_{0j}$'s and $Y_{1j}$'s, while identity activation is applied to continuous $Y_{0j}$'s and $Y_{1j}$'s. We have observed in the simulation experiment in section 4.1 that the representation parameters can be recovered from the model fitting therefore preserving the scientific meanings of the latent variables.

**Loss function for learning latent representations**   We denote $\boldsymbol{\theta}$ as all the parameters in the model. We further denote $f_{0j}(\mathbf{Z}_{i0}; \boldsymbol{\theta})$ as the model fit for $Y_{i0j}$ (the $j$-th item for subject $i$ at baseline), and $f_{1j}(\mathbf{X}_i, \mathbf{Z}_{i0}; \boldsymbol{\theta})$ as the model fit for $Y_{i1j}$. We train this model by minimizing the following objective function $\frac{1}{n} \sum_{i=1}^n w_i \Big\{ \sum_{j=1}^J L\big(f_{0j}(\mathbf{Z}_{i0}; \boldsymbol{\theta}), Y_{i0j}\big) + \sum_{j=1}^J L\big(f_{1j}(\mathbf{X}_i, \mathbf{Z}_{i0}; \boldsymbol{\theta}), Y_{i1j}\big) \Big\}$, where $w_i = \frac{I(A_i=1)}{P(A_i=1|\mathbf{X}_i)} + \frac{I(A_i=-1)}{P(A_i=-1|\mathbf{X}_i)}$, and $L(\cdot, \cdot)$ is a loss function. For continuous $Y_{0j}$ and $Y_{1j}$, we use squared error loss, and for discrete $Y_{0j}$ and $Y_{1j}$ we adopt cross entropy loss. By using these loss functions, this objective function is related to the likelihood function for the observed items, and we optimize with respect to both pre- and post-treatment multi-domain outcomes. We use inverse probability weighting ($w_i$) to account for different treatment group sizes.

**Computational algorithms**   Because the baseline latent state variables $\mathbf{Z}_{i0}$'s are not observed, for model fitting, we iterate between updating $\mathbf{Z}_{i0}$'s for all subjects and updating the model parameters $\boldsymbol{\theta}$ until the objective function converges. Since $\mathbf{Z}_{i0}$ contains binary state variables and usually a low-dimensional $\mathbf{Z}_{i0}$ is found sufficient to represent the key domains of mental disorders [25, 27, 28], we can search through all possible values of $\mathbf{Z}_{i0}$ and update it with the one that yields the smallest loss for subject $i$. The update for $\mathbf{Z}_{i0}$'s can be done in parallel for each individual. Given the current values of $\mathbf{Z}_{i0}$'s, we update $\boldsymbol{\theta}$ using stochastic gradient descent (SGD). Since the update for $\boldsymbol{\theta}$ is approximate and it requires more steps to converge, we update $\boldsymbol{\theta}$ with $M$ gradient descent steps and perform one step of exact search for $\mathbf{Z}_{i0}$'s.

**Optimal treatment recommendation**   The ultimate goal of learning optimal ITRs is to use fitted rules to recommend treatments for future patients. For a new patient, we only observe his/her baseline measures $(\mathbf{y}_0, \mathbf{x})$. In order to recommend the best treatment, we first infer his/her baseline mental states by searching for the ones that yield the smallest loss in $\mathbf{y}_0$ given by $\widehat{\mathbf{z}}_0 = \text{argmin}_{\mathbf{z}_0} \sum_{j=1}^J L\big(f_{0j}(\mathbf{x}, \mathbf{z}_0; \widehat{\boldsymbol{\theta}}), y_{0j}\big)$, where $\widehat{\boldsymbol{\theta}}$ denotes the fitted parameters. We have shown in the simulation experiment in section 4.1 that learning $\mathbf{z}_0$ by minimizing the loss over $\mathbf{y}_0$ only also achieves very good accuracy. Next, we predict the optimal treatment as the one that yields the highest aggregate mental health state $g(\mathbf{z}_1)$ post treatment given by $\widehat{d}(\mathbf{x}, \widehat{\mathbf{z}}_0) = \text{argmax}_a g(h_z^{(a)}(\mathbf{x}, \widehat{\mathbf{z}}_0; \widehat{\boldsymbol{\theta}}))$, where $h_z^{(a)}(\mathbf{X}, \mathbf{Z}_0; \boldsymbol{\theta})$ denotes the function that infers $\mathbf{Z}_1^{(a)}$ in the model.

**Advantages of the proposed method**   1) By incorporating multi-domain outcomes with meaningful representation structures, it overcomes the limitation of existing methods that only a single outcome variable can be used.  2) patients' heterogeneity is accounted for using an economical number of latent variables based on measurement theory (i.e., $K < J$); and the reduced number of states serves as regularization and provides power for learning patient representation with a smaller sample size. 3) The latent variables $\mathbf{Z}$ are de-noised and more reliable representations of patients' mental health status than the observed measurements $\mathbf{Y}$; thus, in the change of environment, the optimal ITRs based on $\mathbf{Z}$ will be more generalizable. 4) The invariant structure between $\mathbf{Y}$'s and $\mathbf{Z}$'s in both pre- and post-treatment phases (shared representation as shown in Figure 1.b) preserves the validity of the latent variables, and allows borrowing strength in learning the representation structure using data from both treatment phases. 5) By connecting latent variables in psychological measurement theory, neural network architecture and deep learning computational tools, flexible models can be fit to represent complex treatment effect mechanism while allowing parts of the model to share parameters.

## 4   Experiments

We study the performance of the proposed method in learning optimal ITRs for mental disorders when multi-domain psychological measurements are observed from randomized trials. We compared with the state-of-the-art methods that address heterogeneous treatment effects [29, 23, 17, 13, 11] but

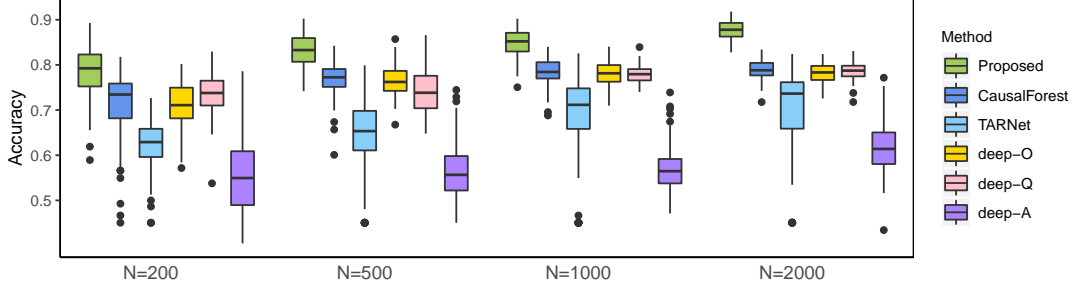

Figure 2: Accuracy of the predicted optimal treatments on the independent test set under the fitted ITRs from 100 simulations with training sample size $N$ of 200, 500, 1000, and 2000.

they can only handle single observed outcome as discussed in section 2. With randomized treatment, the value function defined in section 2 can be expressed in an inverse-probability weighted form, $E\big[R\,I(A = d(\mathbf{X}, \mathbf{Z}_0))\,/\,P(A|\mathbf{X})\big]$, and the empirical value function is given by $\frac{1}{n}\sum_{i=1}^{n}\big[R_i\,I(A_i = d(\mathbf{X}_i, \mathbf{Z}_{i0}))\,/\,P(A_i|\mathbf{X}_i)\big]$. We evaluate all estimated ITRs by calculating their empirical value functions using an independent test set. This value function reflects the expected potential outcome had all subjects in the test set followed the estimated ITR.

## 4.1 Simulation

We assumed there were three latent domains, and the observed outcomes consisted of nine discrete items and five continuous items. We simulated data where the true optimal treatments are known. The detailed data generating mechanism are included in the supplementary material. We assumed the sum of the latent variables represented the overall mental disorder severity. Thus the true optimal treatment was the one that minimized the expected value of $\mathbf{Z}_1$ sum for each subject.

**Model fitting**   Training datasets of sample size 200, 500, 1000, and 2000 were simulated based on the procedure given in the supplementary material. For the state-of-the-arts [29, 23, 17, 13, 11], sum of $\mathbf{Y}_1$ was treated as the outcome variable and $\mathbf{Y}_0$ and $\mathbf{X}$ were included as feature variables.

For the proposed method, we tuned over the hyper-parameters including the number of hidden layers (ranges from 1 to 3) and hidden units (10 to 30), and number of iterations (1 to 10) through cross-validations on the training data. Model with 2 hidden layers of 20 and 10 units was selected. 6 epochs of SGD was implemented with Adam [9] under a learning rate of 0.1 before 1 step of exact search for $\mathbf{Z}_{i0}$, and we ran in total 6 iterations of the update process. Convergence was observed in all cases. To achieve identifiability of the latent variables, we have controlled their directions and exchangeability in model fitting (details are described in the supplementary material). Essentially, for each latent construct, we set initial values of the loading parameters for one observed item, which is analogous to fixing one loading path per latent variable in factor analysis. In the real-world applications, prior knowledge on the direction of certain observed symptom items and potential latent domains is often available before fitting the model. We built the proposed model using Pytorch. For model fitting, the computing time is 5s, 10s, 35s, and 72s under sample size of 200, 500, 1000, and 2000 with a batch size of 100, 250, 250, and 500 on a 2.7 GHz Intel Core i5 processor under the specified training procedure.

Fitted ITRs from all methods were evaluated on an independent test set of sample size 100,000 simulated under the same setting. We evaluated the fitted ITRs in terms of the prediction accuracy for the optimal treatments, value function for the overall disorder severity (represented by the sum of $\mathbf{Z}_1$), and also the value function for some indicative measures in $\mathbf{Y}_1$, denoted as $\widetilde{Y}_1$. We considered $\widetilde{Y}_1 = Y_{11} + Y_{16} + Y_{17} + Y_{110} + Y_{111}$ since these items are most indicative of the latent domains, and in practice they could be some observed disease symptoms that we particularly care about. In summary, we aimed to minimize the overall disorder severity (represented by the sum of $\mathbf{Z}_1$) and also important observed symptoms (represented by $\widetilde{Y}_1$).

**Simulation results**   We observed adequate fit of the proposed model. The average prediction accuracy was above 0.6 for the multi-category $\mathbf{Y}_0$'s even under training sample size of 200. A higher

Table 1: Value function of the fitted ITRs with respect to the latent disorder severity (sum of $\mathbf{Z}_1$; the lower, the better) and the predictive symptoms (sum of $\widetilde{Y}$; the lower, the better) on the independent test set, mean (standard deviation), from 100 simulations under different training sample size $N$

| $N$ | | Proposed | CausalForest | TARNet | Deep-O | Deep-Q | Deep-A |
|---|---|---|---|---|---|---|---|
| Value function for latent disorder severity (sum of $\mathbf{Z}_1$; the lower, the better) | | | | | | | |
| 200 | Mean (sd) | **1.80** (0.04) | 1.85 (0.06) | 1.93 (0.05) | 1.85 (0.04) | 1.83 (0.04) | 2.00 (0.08) |
| 500 | Mean (sd) | **1.77** (0.02) | 1.80 (0.03) | 1.91 (0.07) | 1.80 (0.02) | 1.80 (0.02) | 1.98 (0.06) |
| 1000 | Mean (sd) | **1.76** (0.01) | 1.79 (0.02) | 1.88 (0.09) | 1.79 (0.01) | 1.79 (0.01) | 1.98 (0.04) |
| 2000 | Mean (sd) | **1.74** (0.01) | 1.79 (0.01) | 1.87 (0.09) | 1.79 (0.01) | 1.79 (0.01) | 1.98 (0.04) |
| Value function for predictive symptoms (sum of $\widetilde{Y}$; the lower, the better) | | | | | | | |
| 200 | Mean (sd) | **12.74** (0.19) | 12.91 (0.28) | 13.30 (0.22) | 12.92 (0.19) | 12.84 (0.17) | 13.62 (0.32) |
| 500 | Mean (sd) | **12.59** (0.10) | 12.70 (0.10) | 13.22 (0.33) | 12.72 (0.09) | 12.71 (0.10) | 13.53 (0.24) |
| 1000 | Mean (sd) | **12.57** (0.09) | 12.65 (0.06) | 13.06 (0.40) | 12.65 (0.06) | 12.67 (0.06) | 13.55 (0.17) |
| 2000 | Mean (sd) | **12.52** (0.05) | 12.62 (0.04) | 13.01 (0.41) | 12.64 (0.06) | 12.67 (0.06) | 13.54 (0.17) |

overall prediction accuracy for $\mathbf{Y}_0$ cannot be achieved because some items were simulated with high noise and low correlation with $\mathbf{Z}_0$, and so the prediction accuracy for those items was expected to be low. On the other hand, for those items with large loadings on $\mathbf{Z}_0$ (e.g., items selected in $\widetilde{Y}_1$), the estimated parameters $\widehat{\boldsymbol{\beta}}_{jk}$ were learned in the correct directions with the estimated values very close to the true values (or the equivalent values for the multinomial distribution). In other words, we can recover the representation structures and therefore preserve the scientific meanings of the latent domains.

Results of the fitted ITRs evaluated on the independent test data from 100 simulations are summarized in Table 1, Figure 2, and Table B.1 in the supplementary material. The proposed method achieves the highest prediction accuracy for the optimal treatments, the lowest disorder severity (sum of $\mathbf{Z}_1$) and the lowest symptom scores $\widetilde{Y}_1$ in all cases with statistical significance ($p$-value $< 0.001$ from pairwise t-tests). Variance of our estimator is also among the smallest. This is favorable because even though our objective function is to minimize the latent disorder severity, the indicative symptoms are simultaneously reduced. When sample size increases, our method yields better mean value with lower variance while the alternative methods barely improve beyond sample size of 500. These results demonstrate the desired property of our method in finding the meaningful representation of the underlying constructs while de-noising the mixed observed measurements.

Furthermore, we obtained the prediction accuracy of 0.956, 0.953, 1.000, and 1.000 for the baseline latent $\mathbf{Z}_0$ on the test set under training sample size of 200, 500, 1000, and 2000. This indicates that even with limited information (only baseline $\mathbf{Y}_0$) from future patients, we are able to learn their latent mental states $\mathbf{Z}_0$ accurately. These results show the benefit of the invariant structure between $\mathbf{Z}$ and $\mathbf{Y}$ pre- and post-treatment, so that one can borrow strength in model training using data from both phases while obtaining high prediction accuracy using only pre-treatment data for future patients.

### 4.2 Application to STAR*D study

**Data and model** STAR*D [22] is a multi-site, multi-level randomized clinical trial designed to compare various treatment regimes for patients with major depressive disorder (MDD). At the first treatment level, all patients were prescribed Citalopram (CIT) as the initial treatment, and those who failed to respond to CIT within 8 weeks were randomized at level 2 within their preference group to switch to a different treatment or to augment treatment by adding another one in addition to CIT.

Our goal was to recommend optimal second-line treatments for those MDD patients who didn't respond to the initial CIT treatment. In order to compare with the state-of-the-art methods for heterogeneous treatment effects, we considered two treatment options: 1) antidepressants that are selective serotonin reputake inhibitors (SSRIs, alone or in combination), and 2) non-SSRI treatments.

To learned optimal ITRs, we included in $\mathbf{Y}_0$ the 17 discrete-valued symptom items in HAM-D, the clinician-rated Quick Inventory of Depressive Symptomatology (QIDS), and the Work and Social Adjustment Scale (WSAS) measured after first level CIT treatment. The corresponding measures

assessed after the level 2 treatments were included in $\mathbf{Y}_1$. Sparse categories "3" and "4" in the HAM-D items were combined to increase the learning power. We included in $\mathbf{X}$ a patient's age, gender, race, family history of depression, and preference of level 2 treatment (switching or augmentation), as well as the Clinical Global Impressions (CGI) Scale and the Global Rating of Side Effect Burden (GRSEB) reported at the end of level 1 treatment. For the comparison methods, the sum score of $\mathbf{Y}_1$ was used as the outcome variable to minimize while $\mathbf{Y}_0$ and $\mathbf{X}$ were included as feature variables. 808 patients from STAR*D were included in the analysis with a mean age of 43 and 59% being females.

In the proposed method, we chose 4 latent mental domains since in the literature of psychiatry, usually 2 to 4 latent factors were identified from models fitted on depression symptoms [27, 28, 25] and for us 4 latent variables yielded the lowest generalization error empirically. We used the same model structure and fitting procedure as in section 4.1. In order to construct a meaningful $g(\mathbf{Z}_1)$ to optimize, we scored each latent domain identified in the model by a value from $-1$ to $1$ so that positive scores were assigned to the "healthy" mental domains, and negative scores to those "unhealthy" domains. Since a higher value in each HAM-D item indicates more severe symptoms and the representation parameters $\boldsymbol{\beta}_{jk}$ reflect the association between item $j$ and latent domain $k$ as discussed in section 3.2, we computed the score for the latent domain $k$ as the proportion of negative trend minus positive trend observed in adjacent entries in $\widehat{\boldsymbol{\beta}}_{jk}$ for all $j$. Therefore, by maximizing the sum of the latent domains after scoring, we equivalently maximized each patient's overall mental health state.

We conducted 4-fold cross validations to evaluate the estimated ITRs (i.e., fit ITRs on 3 folds of data and compute the empirical value function from the remaining fold). We examined performance on four outcome measures post second-line treatments, namely, HAM-D sum score, QIDS, WSAS, and CGI. Lower value in HAM-D and QIDS indicates less depression symptoms; lower value in WSAS indicates better functioning in work and social life; and lower value in CGI indicates greater improvement acquired from treatment rated by the clinicians. Empirical value functions were calculated on the validation sets with respect to these four measures.

Table 2: Value function under the fitted ITRs, mean (standard deviation), evaluated with respect to four post-treatment outcomes (lower values indicate healthier states and thus better performance) from 100 cross validations on the STAR*D data. An asterisk (*) indicates that our method is significantly better using pairwise t-test ($p$-value $< 0.05$).

| Outcome measure | Proposed | CausalForest | TARNet | Deep-O | Deep-Q | Deep-A |
|---|---|---|---|---|---|---|
| HAMD | **12.23 (0.38)** | 12.62 (0.45)* | 12.37 (0.51)* | 13.59 (0.38)* | 13.55 (0.50)* | 12.32 (0.54) |
| QIDS | **8.33 (0.32)** | 8.97 (0.30)* | 8.63 (0.38)* | 9.66 (0.28)* | 9.52 (0.36)* | 8.57 (0.42)* |
| CGI | **2.54 (0.08)** | 2.72 (0.08)* | 2.61 (0.11)* | 2.92 (0.08)* | 2.89 (0.12)* | 2.60 (0.12)* |
| WSAS | 20.27 (0.46) | **19.98 (0.72)** | 20.05 (0.77)* | 22.16 (0.73)* | 21.99 (0.83)* | 20.14 (0.73) |

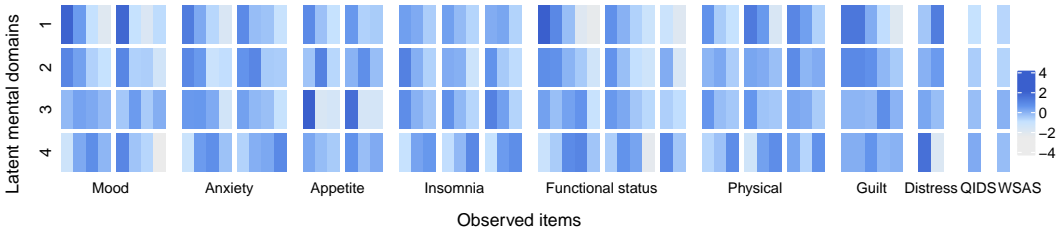

Figure 3: Fitted representation parameters $\widehat{\boldsymbol{\beta}}_{jk}$'s for the model between the 4 latent mental domains and 19 observed item measurements from STAR*D. The first 17 are discrete-valued items in HAM-D where similar items are grouped together (e.g., first 2 items for "Mood" are grouped together). HAM-D: Hamilton Depression Rating Scale, QIDS: Quick Inventory of Depressive Symptomatology, WSAS: Work and Social Adjustment Scale.

**Results** Results from 100 cross validations are summarized in Table 2. The proposed method achieves the lowest mean score in HAMD, QIDS, and CGI, and the second lowest score in WSAS on the validation sets, and pairwise t-tests show superiority of our method with statistical significance

for most of the outcomes. The variability of our method is among the lowest for all outcomes, indicating our method is more stable. These results show the broad utility of the treatment polices we learned: they lead to relief of the depressive symptoms (HAMD, QIDS) and achieve greater clinical improvements (CGI) as well as better work and social functioning (WSAS) at the same time. This demonstrates the utility of the latent representations we learned. They serve as more useful characterization of patients' mental health states than the raw observed measures and lead to better patient outcomes on multiple domains. In comparison, causal forest achieves the lowest score on WSAS, but not on the other three outcomes. TARNet and Deep-A have similar and overall good performance on the four outcomes, while Deep-O and Deep-Q do not perform well on any of the outcomes. These results indicate that the latent outcomes learned in our method are more clinically relevant than the naive sum score used in the comparison methods, and therefore our treatment rules lead to better clinical outcomes.

We plot in Figure 3 the heatmap of the fitted representation parameters $\widehat{\boldsymbol{\beta}}_{jk}$'s for the relation between the 4 latent mental domains and the 19 observed measurements. The 17 discrete-valued items in HAM-D are grouped together (e.g., mood items, anxiety items etc.). For HAM-D item $j$, each value in $\widehat{\boldsymbol{\beta}}_{jk} = (\widehat{\beta}_{jk0}, ..., \widehat{\beta}_{jkl_j})^T$ is plotted as an individual column. Hence, as explained in section 3.2, the color trend across the columns for each HAM-D item indicates its correlation with each mental domain. For example, the three items for insomnia are negatively correlated with latent domain 1,2,3 (darker to lighter color from left to right) and positively associated with latent domain 4 (lighter to darker color from left to right). Similar items in HAM-D (in the same groups in Figure 3) show similar representation structures on the latent domains, which indicates that the learned latent domains preserve the scientific contexts. Additionally, clinical interpretation of the latent mental domains can be made based on $\widehat{\boldsymbol{\beta}}_{jk}$'s. The first domain is characterized by good mood, less guilt feeling, and better functional and physical status; the second domain represents less anxiety and better mood; the third domain represents better appetite and less weight loss; a higher value in the fourth domain indicates higher severity in insomnia and distress.

# 5    Discussion

In this work, to address the challenge of between-patient heterogeneity manifested in multi-domain observed outcomes, we propose an integrated learning framework where patients' underlying health status are meaningfully represented with latent variables and the treatment that optimizes the underlying health status for each individual can be discovered. This representation structure is assumed to be invariant before- and post-treatment based on measurement theory of validated instruments that are used to evaluate treatment outcomes (e.g., HAM-D), and it preserves the meaning of the latent variables for scientific interpretations. Our approach performs better than the baseline methods where a simple summary score is used on simulated data and a real-world clinical trial data in learning individualized treatment polices for mental disorders, and shows broad utilities which lead to better patient outcomes on multiple domains.

Here we demonstrate the proposed method with randomized trials; however, our method can be easily modified and applied to observational studies under the assumption of no unmeasured confounding. This assumption, also known as ignorability, is commonly assumed in learning heterogeneous treatment effects for observational studies [29, 23], and it ensures the identifiability of causal effects conditional on observed confounding variables. To this end, to remove observed confoundings, our algorithm can allow more efficient matching or weighting by using the lower-dimensional latent variables to reduce dimensionality.

Additionally, the proposed method is sufficiently general to provide optimal ITRs for any function of the latent representations. For example, domain-specific treatment rules can be constructed if treatment targets a particular latent mental domain. In this paper, we examined the utility of the proposed method for learning ITRs for major depressive disorder. Future work can be done to validate it on different psychiatric disorders. Since measurement models have been successfully applied to other mental disorders, for example, anxiety [3] and post-traumatic stress disorder (PTSD) [8], we expect our method to perform well on them. Another potential is to extend our method to learn dynamic treatment policies where optimal treatment decisions are inferred at multiple time points for recurrent chronic diseases to improve patients' long-term health outcome.

## Broader impact

Current treatments for mental health disorders are largely inadequate, in part due to the extensive heterogeneity between patients. This work may improve treatment responses for patients with mental disorders by facilitating clinical decision makings in prescribing personalized treatments depending on patient-specific measures. We note that it is possible that the latent constructs derived from poor instruments can have bias in certain subpopulations. Therefore, the psychometric measures to be included in the model need to be carefully studied including their reliability, validity and differential item functioning or item-response bias in populations of diverse racial or socioeconomic factors. For best practice, the latent constructs should be validated on external studies or among different subgroups. Our application uses carefully chosen instruments, which have been shown to exhibit adequate psychometric properties and widely used in clinical settings. Failure in recommending optimal treatments in certain populations may lead to patients' sub-optimal treatment responses. However, our research serves as a recommendation tool to assist clinical decision making.

## Acknowledgments and Disclosure of Funding

This research is supported by U.S. NIH grants MH117458, NS082062 and NS073671.

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
