[Supplementary Material]

# Supplementary Material for

# Representation Learning for Optimal Individualized Treatments with Multivariate Outcomes

**Yuan Chen**
Department of Biostatistics
Columbia University
New York, NY 10032
yc3281@cumc.columbia.edu

**Donglin Zeng**
Department of Biostatistics
University of North Carolina at Chapel Hill
Chapel Hill, NC 27516
dzeng@email.unc.edu

**Tianchen Xu**
Department of Biostatistics
Columbia University
New York, NY 10032
tx2155@cumc.columbia.edu

**Yuanjia Wang**
Department of Biostatistics
Columbia University
New York, NY 10032
yw2016@cumc.columbia.edu

In this supplementary material, we describe in details the simulation procedures including all parameters, additional model fitting details, and additional simulation results for section 4.1 in the main paper. We also describe the way of scoring each latent domains for the real data application in section 4.2 in the main paper.

## A Simulation procedures and additional model fitting details

In this section, we describe the data generating mechanism in section 4.1 of the main paper. We assumed there were three latent domains, and the observed outcomes consisted of nine discrete items and five continuous items, with three discrete items taking values in $\{0, 1\}$, three taking values in $\{0, 1, 2\}$, and three taking values in $\{0, 1, 2, 3\}$. Each latent domain $Z_{0k}$ was simulated from Binomial(1, 0.5), and each discrete item $Y_{0j}$ was simulated with softmax transformation from $\mathbf{Z}_0$, while continuous $Y_{0j}$ was simulated by a linear transformation of $\mathbf{Z}_0$. Two other baseline features in $\mathbf{X}$ were simulated from $\mathcal{N}(0, 1)$, and treatment assignment $A$ was simulated taking value of 1 or $-1$ with equal probability. Next, each post-treatment latent outcome $Z_{1k}$ was simulated based on $P(Z_{1k} = 1 | \mathbf{Z}_0, \mathbf{X}, A) = \sigma(\theta_{0k} + \boldsymbol{\theta}_{1k}^T \mathbf{Z}_0 + \boldsymbol{\theta}_{2k}^T \mathbf{X} + \theta_{3k} A + \boldsymbol{\theta}_{4k}^T A \mathbf{Z}_0 + \boldsymbol{\theta}_{5k}^T A \mathbf{X})$, where $\sigma(x) = 1/(1 + e^{-x})$. Lastly, $\mathbf{Y}_1$, the observed outcome responses after treatment were simulated based on $\mathbf{Z}_1$ and the same conditional distribution of $\mathbf{Y}_0$ given $\mathbf{Z}_0$. The simulation parameters were given in Table A.1 and A.2.

In order to learn the three latent domains in the correct directions, we control the direction of the estimated parameters for one item per latent domain. Specifically, since $Y_{01}$, $Y_{06}$, and $Y_{07}$ were simulated to be positively correlated with the first, second and the third latent domain respectively; we set the starting value of $\boldsymbol{\beta}_{11}$, $\boldsymbol{\beta}_{62}$, and $\boldsymbol{\beta}_{73}$ to some big positive values $(0, 5)$, $(0, 5, 10)$, and $(0, 5, 8, 10)$. This reflects the real-world situation when we had some prior clinical knowledge about the direction of certain observed symptom items and potential latent domains before fitting the model.

Table A.1: Simulation parameters for the conditional distributions of observed items $Y_{0j}$ given latent domains $\mathbf{Z}_0$ (same for $Y_{1j}$ given $\mathbf{Z}_1$). $Y_{0j}$ is discrete for $j = 1, ..., 9$ and is continuous for $j = 10, ..., 14$.

| j | $\boldsymbol{\alpha}_j$ | $\boldsymbol{\beta}_{j1}$ | $\boldsymbol{\beta}_{j2}$ | $\boldsymbol{\beta}_{j3}$ |
|---|---|---|---|---|
| 1 | (0, -1) | (0, 3) | (0, 1) | (0, 2) |
| 2 | (0, -0.5) | (0, 1) | (0, -1) | (0, 0) |
| 3 | (0, -1) | (0, 0) | (0, 0) | (0, 0) |
| 4 | (0, -1, -1) | (0, 1, 1) | (0, -1, -2) | (0, 0, 0) |
| 5 | (0, -0.5, -1) | (0, 0, 0) | (0, 0, 0) | (0, 0, 0) |
| 6 | (0, -1, -1) | (0, 0, 1) | (0, 2, 4) | (0, 1, 2) |
| 7 | (0, 0.5, -0.5, -1) | (0, 0, 1, 1.5) | (0, 0, 0, 1) | (0, 1, 2, 3) |
| 8 | (0, -0.5, -1, 0.5) | (0, 0, 0, 0) | (0, 2, 1, 0) | (0, 1, -1, -2) |
| 9 | (0, 0.5, -1, -2) | (0, 2, 0, -2) | (0, 0, 0, 0) | (0, 0, 0, -2) |
| 10 | 1 | 1 | 2 | 1 |
| 11 | 2 | 1 | 2 | 2 |
| 12 | 0 | 0 | -2 | 0 |
| 13 | -1 | -1 | 0 | -1 |
| 14 | -2 | -2 | -1 | 1 |

Table A.2: Simulation parameters for the conditional distributions of $\mathbf{Z}_1$ given $\mathbf{Z}_0$, $\mathbf{X}$ and $A$.

| k | $\theta_{0k}$ | $\boldsymbol{\theta}_{1k}$ | $\boldsymbol{\theta}_{2k}$ | $\theta_{3k}$ | $\boldsymbol{\theta}_{4k}$ | $\boldsymbol{\theta}_{5k}$ |
|---|---|---|---|---|---|---|
| 1 | 2 | (1, 0, 0) | (1, 0.5) | -0.5 | (1, -0.5, 0) | (2, -1) |
| 2 | -1 | (0.5, 1, 0.5) | (0.5, -1) | -0.5 | (0, 3, -2) | (-0.5, 2) |
| 3 | 1 | (0, 0, 2) | (0, 0) | 1 | (-2, -1, 2) | (1, 1) |

# B  Additional Simulation Results

We present the simulation result for the accuracy of the fitted ITRs on the independent test set in Table B.1 below, which corresponds to the Figure 3 in the paper. Our methods yields the highest accuracy and the variance is among the smallest among all methods. This result indicates that our method effectively denoises the mixed observed measurements and recovers the true underlying constructs.

Table B.1: Accuracy of the fitted optimal treatment on the test set from 100 simulations for training sample size of 200, 500, 1000, and 2000

| $N$ | | Proposed | CausalForest | TARNet | Deep-O | Deep-Q | Deep-A |
|---|---|---|---|---|---|---|---|
| 200 | Mean | **0.78** | 0.71 | 0.62 | 0.71 | 0.73 | 0.55 |
| | Sd | 0.06 | 0.07 | 0.06 | 0.05 | 0.05 | 0.08 |
| 500 | Mean | **0.83** | 0.77 | 0.64 | 0.77 | 0.77 | 0.57 |
| | Sd | 0.04 | 0.04 | 0.08 | 0.03 | 0.03 | 0.06 |
| 1000 | Mean | **0.85** | 0.78 | 0.68 | 0.78 | 0.78 | 0.56 |
| | Sd | 0.03 | 0.03 | 0.10 | 0.02 | 0.02 | 0.04 |
| 2000 | Mean | **0.88** | 0.79 | 0.69 | 0.78 | 0.77 | 0.57 |
| | Sd | 0.02 | 0.02 | 0.10 | 0.02 | 0.02 | 0.04 |