[Reviews · NeurIPS 2020]

Review 1

Summary and Contributions: In this paper, the authors proposed a learning framework utilizing outcomes and biological measures to learn the patient's underlying mental states for treatments recommendation. The proposed method was evaluated on simulated data and real-world randomized controlled trial data.

Strengths: - The proposed method has several advantages, such as the utilization of multi-domain data, modeling of patients’ heterogeneity, cleaner and more reliable representations of patients’ mental health status, and lastly, invariance representation before and after-treatment, enabling doubling the amount of data for training. - The authors evaluated the proposed method on both a synthetic dataset and a real world clinical trial data demonstrating the effectiveness of the method. Comparisons with other related methods were also conducted. - Availability of code for the reproduction of the experiments.

Weaknesses: The work is quite well done, with a nice derivation of the proposed method and evaluation on synthetic and real-world datasets. To nitpick, it will be interesting to dive into the mental state representation and understand what the representation means.

Correctness: Yes, claims and methods seem to be correct.

Clarity: Yes, the paper is well written.

Relation to Prior Work: Yes, related works are described in the "Related work" section and are compared in the experiments.

Reproducibility: Yes

Additional Feedback:


Review 2

Summary and Contributions: The paper proposes a novel approach for evaluation of individualized treatment rules for mental disorders. The gist of the approach is to model the observed symptoms using latent space variables reflecting an underlying mental state, and the transition of that state under different treatment options. The latent state to symptoms maps are time invariant. The approach is tested on the data for the randomized treatment trial for mental disorders and shows superior results compared to the existing models.

Strengths: The paper proposes a novel model for predicting treatment effects with a latent space relating the observed symptoms. Predicted scores are calculated on the end-intervention symptoms predicted through the latent space transition model. A clear presentation of the model and its comparison to the existing treatment models. Suitable for NIPS, though the novelty of the model in context of general ML research and existing learning algorithms is limited.

Weaknesses: - Scope of the paper is limited to randomized trials. The approach tailored to work on randomized trials data where initial and end states for each intervention are clearly defined. Confounding and treatment group matching is not considered. This should be stressed as one of the limitations of the approach for estimating the treatments effects. The model and the algorithms for learning the model are straightforward and not surprising. The benefit and impact is on the application side. It would be good if the authors added some text on how their model can be used in other then clinical data domain.

Correctness: Appears correct. Develops models for treatment effects and machine learning solutions/algorithms tailored for these models.

Clarity: Yes.

Relation to Prior Work: The paper reviews the majority of existing methods for modeling treatment effects and compares the proposed method to these experimentally. But, it should also put the method in context of existing ML work on latent space transition models and general latent space models.

Reproducibility: Yes

Additional Feedback: Post rebuttal comments: The authors gave reasonable answers to most of the questions/points raised in the reviews. Because of a somewhat limited novelty on the ML methodology side I would like to keep my review at the marginal accept.


Review 3

Summary and Contributions: This paper proposes a method for learning an optimal treatment policy when the target outcome is a function of latent constructs, in this case, latent psychological constructs. The method starts by simultaneously estimating the conditional distribution of latent variables given the confounders and treatment variable. Next, given some observed features and measurements, the MAP latent constructs are inferred. Finally, treatment is chosen to maximize a predefined function of the MAP latent variables. This methodology is tested using synthetic and real data.

Strengths: Optimizing treatment when the true outcome of interest is latent is an extremely challenging problem that I think is highly relevant to the NeurIPS community. Further, I think the high level ideas proposed in this paper are a step in the right direction. In particular, I think that the idea of utilizing knowledge about the causal structure to learn the latent constructs is an excellent idea.

Weaknesses: I have two major concerns about the paper: 1. The premise of the paper is that practitioners would like to optimize a predefined function of latent patient state; however, such a function cannot be predefined since the learned latent constructs do not, a priori, have meaning. More specifically, latent psychological constructs learned by a model such as factor analysis only take on meaning *after* practitioners interpret the learned factors in the context of the underlying items. Further, latent variables may be subject to many different identifiability issues that make defining the function g impossible without first examining the learned latent factors. For example, the proposed model cannot distinguish between Z and 1-Z which would have complementary interpretations. To see this, simply observe that \alpha + \sum_k \beta_k Z_k = \alpha' + \sum_k \beta'_k U_k, where U_k = 1 - Z_k, \alpha' = \alpha + \sum_k \beta_k, and \beta'_k = -\beta_k. 2. More broadly, I have serious concerns about the potential risks of optimizing treatment with respect to learned latent constructs without first extensively validating those constructs. As I'm sure the authors are aware, there is a substantial body of literature on validating psychological measure in order to ensure that they are measuring what is expected. In particular, in the broader impact statement, the authors state "No individual may be put at disadvantage from this research"; however, the authors absolutely cannot guarantee that latent factors learned using the proposed method do not have biases. Latent constructs such as IQ are well-known to have racial and socio-economic biases and to use such measures to optimize treatment before testing for such biases may lead to biased treatment.

Correctness: The paper appears technically correct.

Clarity: I found the description of the methodology hard to follow. I recommend modifying section 3 to start with an overview or pseudocode description of the main steps is the method. Something like: 1. Specify a latent variable causal model describing the relationship between the treatment, latent constructs, and measurements. 2. Estimate the parameters of the model using a hard EM style procedure. This gives us a mapping, m, from pre-treatment measurements and treatment a treatment value to post-treatment latent constructs. 3. For a pre-specified function of the post-treatment latent constructs, g, estimate the optimal treatment policy as the policy that maximizes g(m(Y_0,X,a)). Then structure the description of the method around each of these pieces: model structure, estimation method, policy optimization.

Relation to Prior Work: I thought the discussion of related work was missing any description the relevant methods from psychometrics. In particular, could the authors please describe in more detail how this approach is related to/distinct from first using a latent variable structural equation model to estimate latent constructs (as is standard in psychology) and then choosing treatment based on those constructs?

Reproducibility: No

Additional Feedback: --- Post discussion phase --- After discussion with the other reviewers and reading the author response, I have decided to keep my score at a 4. In particular, I do not think the reviewers have adequately addressed the concerns related to potentially harmful bias encoded in the method. Specifically, I would like to respond to the following points made in the author response: 1) The authors claim that because the individual items in HAMD have been sufficiently validated, the risk of unintentional bias is low. This may be true in the case of HAMD (and it is laudable that the authors consulted psychologists regarding the resulting measures), but the method is presented as general, not just for HAMD. Further, items often cannot be evaluated in isolation from the way they are aggregated. For example, the degree (and even direction) of racial bias in IQ tests can be changed by changing the weight given to various questions and sections. 2) The authors argue in their response that "We have empirically shown that our latent constructs lead to improved value function evaluated by other external outcomes not used in training (Table 2)". The value function is an expected reward and does not reflect potential disparities. I feel that the literature on health disparities and algorithmic fairness is pretty clear on this point. 3) Finally, the authors respond that "the risk of such bias is not unique to our method". This is absolutely true, but by removing the typical validation step, I believe the risk of such bias increases.


Review 4

Summary and Contributions: The paper presents a model for latent-space representation of mental state integrated in a predictive multi-domain outcome model using deep neural networks. The model is then used to learn optimal individualized treatment rules (ITR). The authors compared the model against 5 other methods in two experiments using simulated data and data from a randomized clinical trial. The results show evidence to support the model superiority compared to previously suggested models in the field domain.

Strengths: The motivation and objectives of this study are well presented. The theoretical grounding and empirical evaluation in the study are adequate. The work presents a fair novelty to the field, building on previous work by presenting a framework incorporating multi-domain outcomes. The implementation of the model preserves principles from the psychiatry and information theory and provides a cross-field work and is relevant to the NeurIPS community.

Weaknesses: Information on the number of patients in the clinical trial and data distribution in the experiments is missing. Further experimental analysis on different psychiatric disorders is necessary to validate the model, while this might be out of the scope for this work to present these analysis, the authors should relate to the future studies that could lead to a practical and validated applicable model in the future.

Correctness: The claims and methods presented in this work are correct.

Clarity: The paper is communicated with clarity.

Relation to Prior Work: Relation to previous work is well established throughout the paper. The paragraph on advantages of the proposed method is particularly useful.

Reproducibility: Yes

Additional Feedback: Please add the abbreviations and more details on the performance scores in the STAR*D experiment (i.e. HAM-D, QIDS etc.). Since these are not typical machine-learning performance measures and are taken from the domain, it’s important to present the readers with a short explanation to gain better intuition to interpret the results.

[Author Response · NeurIPS 2020]

We thank the reviewers for the constructive comments and suggestions. Our responses are detailed in the following.

**Reviewer 1:** We will add the interpretations of the latent domains. The first domain is characterized by good mood, less guilt feeling, and better functional and physical status; the second domain is mainly depicted by less anxiety and better mood; the third domain is characterized by better appetite and less weight loss; higher value in the fourth domain indicates higher severity in insomnia and distress. We will add more discussions in broader impact, e.g., potential applications to observation studies, behavioral sciences and econometrics to learn effective personalized interventions.

**Reviewer 2:** In this work, we demonstrated the proposed method with randomized trials; however, our method can be easily modified and applied to observational studies under the assumption of no unmeasured confounding. This assumption, also known as ignorability, is commonly assumed in learning heterogeneous treatment effects for observational studies, and it ensures the identifiability of causal effects conditional on observed confounding variables $\mathbf{X}$ and $\mathbf{Y}_0$. Since $\mathbf{X}$ and $\mathbf{Y}_0$ are included and adjusted in our model, their confounding effects can be removed. Additionally as suggested, matching or weighting using lower-dimensional learned latent $\mathbf{Z}_0$ can better remove confounding when $\mathbf{Y}_0$ is high-dimensional. Thus, our algorithm allows more efficient matching due to the reduced dimensionality in feature variables by using the lower-dimensional latent variables.

To our best knowledge, our method is the first to incorporate multi-domain outcomes as opposed to a single scalar outcome in learning individualized treatment rules (ITRs), and we combine measurement models and ITR learning under an unified loss where the latent states and ITRs are simultaneously learned with an iterative procedure. The proposed model is very general; it can be applied to other areas where reward outcome is latent and needs to be inferred from many measurements, e.g., behavioral sciences and consumer preference in marketing for effective interventions. Additionally as suggested, we will add discussions in related work on latent variable models, e.g., hidden Markov models. Since they only estimate latent states but do not consider learning treatment rules, one straightforward approach is to conduct a two-step procedure. But the two-step procedure will not be as efficient as our approach which borrows strength from data before and after treatment with shared representation parameters while preserving the consistency of the latent constructs.

**Reviewer 3:** The choice of the predefined $g()$ function of the latent states may depend on application but a common choice is the total sum similar to aggregating sub-scales of instruments commonly used in psychiatry [3]. We described in Section 4.2 (starting from line 283 and Supplementary material Section C) about how to ensure $g(\mathbf{Z}_1)$ is meaningful to optimize. In summary, based on the loading parameters, the latent constructs are interpretable and we scored each latent domain so that their directions are aligned, which ensures that their sum score is meaningful to optimize. To achieve identifiability of the latent variables, we have controlled their directions and exchangeability in model fitting (Section 4.1 starting from line 225 and Supplementary Section A). Essentially, for each latent construct we set initial values of the loading parameters for one observed item, which is analogous to fixing one loading path per latent variable in factor analysis. Identifiability of the latent constructs was achieved under this value initialization: loading parameters in $P(\mathbf{Y}_0 \mid \mathbf{Z}_0)$ were recovered and we obtained a prediction accuracy of 100% for the latent $\mathbf{Z}_0$ under training sample size of 1000 and 2000 (described in Section 4.1 Simulation results, in the last paragraph).

The validity of the latent constructs is important but not a serious concern in our application because the instrument (HAMD) we used to infer latent states have already been validated, shown to exhibit adequate psychometric properties [2] and widely used in clinical settings. The latent constructs are the lower-dimensional representations of these instruments. We have empirically shown that our latent constructs lead to improved value function evaluated by other external outcomes not used in training (Table 2). In addition, we consulted psychiatrists on the interpretations of the latent constructs and will discuss validation on external datasets in the paper. For the broader impact, it is possible that the latent constructs from poor instruments can have certain bias in terms of their correlations with racial or social-economic factors. However, the risk of such bias is not unique to our method, and our application uses carefully chosen instruments to minimize potential bias. As suggested, we will discuss this point in the Broader impact.

For related work, we will add relevant methods from psychometrics as suggested. In fact, although our model for $P(\mathbf{Y}_0|\mathbf{Z}_0)$ and $P(\mathbf{Y}_1|\mathbf{Z}_1)$ is similar to measurement models with latent constructs, our method is fundamentally different from the traditional two-step procedure where latent constructs are estimated first and then ITRs based on them are identified. In contrast, we adopt a unified approach where the latent constructs and ITRs are simultaneously learned. This is more efficient and borrows strength from data before and after treatment with shared representation parameters which preserves the validity of the latent constructs. About clarity, since our method is a simultaneous approach, we introduced the key concepts for ITR first and then break down to each component of the model. We will add a paragraph to guide reading. For reproducibility, we have submitted the codes for reproducing all experiments as Supplementary and all necessary details for reproducing the results are presented in the paper and Supplementary.

**Reviewer 4:** 808 patients from STAR*D were included (Section 4.2 under Data and model). We will provide more information about data distribution, e.g., mean age is 43, with 59% females. As suggested, we will discuss validating on different psychiatric disorders as future work. Since measurement models have been successfully applied to other mental disorders [1], we expect our method to perform well on them. Abbreviations and short descriptions for the performance scores are provided in Section 4.2 under Data and model. We will include them in Figure 3 for the ease of reading.

[1] E. S. Barratt. Factor analysis of some psychometric measures of impulsiveness and anxiety. *Psychological reports*, 16(2):547–554, 1965.

[2] G. A. Fava, R. Kellner, F. Munari, and L. Pavan. The hamilton depression rating scale in normals and depressives. *Acta Psychiatrica Scandinavica*, 66(1):26–32, 1982.

[3] N. Rose, W. Wagner, A. Mayer, and B. Nagengast. Model-based manifest and latent composite scores in structural equation models. *Collabra: Psychology*, 5(1), 2019.


[Meta-Review · NeurIPS 2020]

The motivation and objectives of this study are well presented. The theoretical grounding and empirical evaluation in the study are adequate. The work presents a fair novelty to the field, building on previous work by presenting a framework incorporating multi-domain outcomes. The implementation of the model preserves principles from the psychiatry and information theory and provides a cross-field work and is relevant to the NeurIPS community. There is a criticism that needs to be addressed in the discussion in the final version is how sensitive the method is to biases and what could be done to avoid the biases when using the method. NOTE FROM PROGRAM CHAIRS: The paper is accepted, however please revise and expand the Broader Impact statement in the camera-ready version. As one reviewer writes, "Latent psychological constructs are well known to be rife with bias and piping them directly into treatment policy decision making without first validating them presents the risk of propagating that bias into treatment decisions." This risk, as well as possible mitigations, should be discussed in the impact section especially given the sensitive subject matter (mental health).